# Immunonutritional Bioactives from *Chenopodium quinoa* and *Salvia hispanica* L. Flour Positively Modulate Insulin Resistance and Preserve Alterations in Peripheral Myeloid Population

**DOI:** 10.3390/nu13051537

**Published:** 2021-05-02

**Authors:** Raquel Selma-Gracia, Polona Megušar, Claudia Monika Haros, José Moisés Laparra Llopis

**Affiliations:** 1Molecular Immunonutrition Group, Madrid Institute for Advanced Studies in Food (IMDEA-Food), Ctra. de, Canto Blanco, n°8, 28049 Madrid, Spain; raquelselgra@iata.csic.es (R.S.-G.); polona.megusar@gmail.com (P.M.); 2Instituto de Agroquímica y Tecnología de Alimentos (IATA), Consejo Superior de Investigaciones Científicas (CSIC), Av. Agustín Escardino 7, Parque Científico, Paterna, 46980 Valencia, Spain; cmharos@iata.csic.es; 3Department of Food Science, Biotechnical Faculty, University of Ljubljana, 1000 Ljubljana, Slovenia

**Keywords:** insulin resistance, innate immunity, myeloid population, obesity

## Abstract

Innate immunity plays a determinant role in high fat diet (HFD)-induced insulin resistance. This study compares the effects of immunonutritional bioactives from *Chenopodium quinoa* (WQ) or *Salvia hispanica* L. (Ch) when used to partially replace wheat flour (WB) into bread formulations. These flours were chosen to condition starch and lipid content in the products as well as because their immunonutritional activity. To be administered with different bread formulations, HFD-fed C57BL/6J mice were distributed in different groups: (i) wild type, (ii) displaying inherited disturbances in glucose homeostasis, and (iii) displaying dietary iron-mediated impairment of the innate immune TLR4/TRAM/TRIF pathway. We analyze the effects of the products on glycaemia and insulin resistance (HOMA-IR), plasmatic triglycerides, intestinal and hepatic gene expression and variations of myeloid (MY), and lymphoid (LY) cells population in peripheral blood. Our results show that feeding animals with WQ and Ch formulations influenced the expression of lipogenic and coronary risk markers, thus attaining a better control of hepatic lipid accumulation. WQ and Ch products also improved glucose homeostasis compared to WB, normalizing the HOMA-IR in animals with an altered glucose and lipid metabolism. These positive effects were associated with positive variations in the peripheral myeloid cells population.

## 1. Introduction

Nowadays, more than 1600 million people worldwide are overweight or obese. According to the World Health Organization (WHO), this global health problem affects all groups of the population and this number will continue increasing by >140% in 2050. Obesity, together with other chronic diseases (i.e., type 2 diabetes–T2D, hypertension, dyslipidemia, physical inactivity, chronic kidney and liver diseases, and smoking), contributes an additional risk factor to cardiometabolic health, while hyperglycaemia confers a substantial independent risk for the adverse physiological outcomes [1]. For example, gestational hyperglycaemia and obesity are independently associated with adverse outcomes during pregnancy [1,2]. Worldwide hyperglycaemia kills some 3.4 million people a year (WHO). Beyond alleviating glucose levels, the greatest benefits for disease prevention appear to be derived from improving the ‘glycemic control’, while the normalization of glycaemia results in a more modest reduction of the effects [3].

Non-alcoholic fatty liver disease (NAFLD) incurs a high risk for the development of T2D and other major features of the metabolic syndrome [4]. Metabolic imbalances, including those affecting glucose homeostasis, arise from complex interactions between genetic and environmental factors [5]. While the host’s endogenous factors are rather difficult to influence, the environmental (i.e., dietary) factors are predominant and addressable from a preventive or therapeutic approach. Besides, emerging evidence supports the metabolic (re)programming of defined components of the innate immune system (i.e., innate immune both myeloid and lymphoid cells), which act as key mediators to induce obesity [6,7]. Furthermore, significant changes in the subpopulations of lymphocytes have been reported in young adults with metabolic syndrome [8]. Notably, intestinal innate immune ‘Toll-like’ receptor (TLR)-4 is a well-known element, which has been identified as an essential regulator of insulin resistance and accumulation of macrophages within the adipose tissue [5]. These immune changes are critical mechanism(s) to normalize the distinctive stamp of obesity in the glucose homeostasis dysregulation. In this context, immunonutritional strategies can play an important role in controlling the severity and progression of the disease. However, whether the use of bread formulation prepared by replacing wheat flour [9] modulates myeloid and lymphoid populations and glycaemic control remains elusive.

The question of how endogenous (i.e., transgenerational inheritance) [10] and environmental factors (i.e., diet) [11] influence early stages and the onset of alterations in lipid and glucose homeostasis has promoted the development of different preclinical models. The exposure of non-pregnant female mice to the obesogenic tributyltin (TBT) enabled a transgenerational inheritance of disturbances in glucose homeostasis [10] and inhibition of the insulin receptor expression [12]. Furthermore, it caused a permanently metabolic (re)programming towards the hepatic fat accumulation and weight gain, particularly in conditions of caloric excess [10]. Besides, it was demonstrated that there was a development of hypoglycemia associated with hypertriglyceridemia and a transitory insulin resistance in animals fed a high fat diet (HFD) [11]. These long term (56 weeks) preclinical models of T2D revealed that an elevated hyperglycaemia associates with early insulin resistance after 2 weeks under HFD feeding. Metabolic alterations caused by HFD that result in and are associated with hepatic steatosis implies a dysmetabolic hepatocellular iron uptake [13]. In addition to this, HFD-induced activation of the TLR4/MyD88 pathway leads to an inhibited macrophage proliferation, which associates with macrophage infiltration into adipose tissue [14]. The TLR4/MyD88-dependent signaling likely contributes to promote hepatic inflammation [15], whereas the production of type I interferons via TLR4/TRIF-dependent signaling appears to exert protective roles in the metabolic dysfunction [16]. This raises interest in evaluating to what extent modulating the immunonutritional potential of food formulations can influence changes in the proportions of peripheral myeloid immune cells in individuals with alterations in the glucose homeostasis.

Bread formulations enriched with *Chenopodium quinoa* and *Salvia hispanica* L. flours provide an effective alternative to improving metabolic imbalances derived from a HFD intake in hyperglycaemic mice [9]. These effects were attributed to immunonutritional bioactives (protease inhibitors found in the low molecular weight albumins/globulin fractions) that interact with TLR4 [17,18,19,20]. Notwithstanding, this interaction appears different to that exerted by immunonutritional bioactives from wheat [21]. At the molecular level, compounds derived from *C. quinoa* and *Salvia hispanica* L. display glycoside and glucuronide groups bound to the amino acid backbone, while those from wheat (*Triticum durum*) lack glycosidic prosthetic groups [17]. Proteome analyses on human-like macrophages shed some light on the biological activity of protease inhibitors from *C. quinoa* and *S. hispanica* L, supporting their correlation with TLR4/TRIF signaling [19] that contrasts with TLR4/MyD88 of wheat [21].

In this study, we hypothesize that partial replacement of wheat flour by that of *C. quinoa* and *S. hispanica* L. will improve the immunonutritional potential of bread formulations. More specifically, such substitution will positively contribute to ameliorate alterations of glucose and lipid homeostasis through the modulation of the innate immune potential within the ‘gut-liver’ axis. To this end, different preclinical models that reproduce major features of the obesogenic and hyperglycemic conditions mimicking the human disease are used. These results can provide novel points of view with significant implications for the ongoing debate about developing appropriate interventions to maximize the beneficial effects and immunometabolic control of obesity.

## 2. Materials and Methods

### 2.1. Bread Samples

Distinct bread formulations were prepared containing different proportions of flour from white quinoa (WQ) at 25% and chia (Ch) at 20% and were compared to wheat bread (WB) [22]. The different proportions of *C. quinoa* and *S. hispanica* flour were chosen to normalize the protein content in the products because as immunonutritional bioactives derive from it. The chemical composition of *C. quinoa*- and *S. hispanica* L-containing bread formulations is shown in Table 1.

### 2.2. Animals

C57BL/6 mice with 6 weeks of age were obtained from the Centro de Investigaciones Biológicas (CIB-CSIC) in Madrid, Spain. Animal experiments were carried out in strict accordance with the recommendations in the Guide for the Care and Use of Laboratory Animals of CSIC (Consejo Superior de Investigaciones Científicas) and the protocol was approved by its Ethics Committee (Proex No.080/19). Mice were maintained under a controlled environment of temperature (21–23 °C) and humidity (55%) and a 12 h:12 h (light:dark) cycle with food and water ad libitum. After treatment, mice were sacrificed by cervical dislocation.

### 2.3. Experimental Design

Different models were used to reproduce the main metabolic and immunological alterations in the functionality of the “gut-liver” axis and were of great relevance in the influence of glucose homeostasis (Figure 1). All bread formulations were administered (14 mg/day/animal) to the different animal models; three times per week for 3 weeks to model 1 (Figure 1A) and for 2 consecutive days to model 2 (Figure 1B).

The amount of bread administered was established according to the daily nutritional recommendation for bread consumption (i.e., 150 g/day/70 kg body weight) that was previously proved to be effective controlling glucose homeostasis [9]. Food intake and changes in body weight of each group were monitored every 2 days. Livers were removed and weighed to calculate the whole body to liver weight (hepatosomatic index). After treatment, mice were killed by cervical dislocation. Animals put on a HFD but not receiving bread formulations were used as controls.

### 2.4. Biochemical Parameters

Blood samples were centrifuged (6000× *g*/10 min) to get clear supernatants. Glucose, insulin, and triglycerides concentrations were determined in plasma samples. Glucose was determined by glucose kit (KA1648, Abnova, Taoyuan, Taiwan), insulin by ELISA kit (RAB0817-1KT, Sigma-Aldrich, Darmstadt, GER) and triglycerides with a commercial kit (Cayman) (n° 10010303). The Homeostatic Model Assessment of Insulin Resistance (HOMA-IR) value was used to define insulin resistance according to the following formula: [(insulin (μIU/mL)* glycaemia (mg/dL)]/405 [28].

### 2.5. Hemogram

Complete blood count was performed on an automated hemocytometer (Abacus Junior Vet, ELECTROMEDINTER SL) to calculate total red blood cells (RBC) and leukocyte counts (WBC) as well as the lymphocytes (LY), myeloid (MY) (macrophages, monocytes), and granulocytes (GR) (granulocytes, eosinophils, neutrophils) population percentages.

### 2.6. Transcripts of Hepatic Lipogenic and Coronary Risk Markers and Macrophage Identifiers

Validated Gene Expression Assays for murine fatty acid synthase (FASN) (forward 5′-TTC CCA CCA AGT GTG GGT AT -3′, reverse 5′-TGG GAC CTT CAG CTT GCT TC -3′), sterol regulatory element-binding protein 1 (SREBP1a) (forward 5′- TCA AAA CCG CTG TGT CCA GT -3′, reverse 5′- GAC GTC TCA ACC CGC TAG G -3′), prostaglandin-endoperoxide synthase 2 (PTGS2) (forward 5′- AAA AGA GAA CGT GAG AGG GCA -3′, reverse 5′- TCA AAC TGG GAA CGG GTG AC -3′), arachidonate 15-lipoxygenase (ALOX15) (forward 5′- TCC CAT TCT AGG GGA GAG GG -3′, reverse 5′- CCT TGA CCA GCT CAG TAG GC -3′), CD68 (forward 5′- AGA AGT GCA ATG GTG GGT CT-3′, reverse 5′- TGG GGC TTA AAG AGG GCA AG -3′), CD206 (forward 5′- TGC AAG CTT GTA GGA AGG AGG -3′, reverse 5′- GAT TAG AGT GGT GAG CAG GC -3′) and β-actin (forward 5′- GGC TCC TAG CAC CAT GAA GAT CAA -3′, reverse 5′- AGC TCA GTA ACA GTC CGC CTA GAA -3′) was purchased from Applied Biosystems (Foqter City, CA, USA). Qrt-PCR was performed with 500 ng of cDNA from liver sections, using the Universal PCR Master Mix (Applied Biosystems, Foster, CA, USA) Quantitative values were calculated by using the 2^−ΔCt^ method [9].

### 2.7. Statistical Analyses

The results are presented as the mean and standard error of the mean (SEM). The statistical analysis between the different groups of treatment within a same experimental model was conducted using one-way analysis of variance (ANOVA) and the Kruskal–Wallis post hoc test by ranks. Because of the different conditions used to obtain the animals for the different models, the comparison between groups of animals receiving the same bread formulation was performed by comparison of the means. Statistical analyses were performed with the software Statgraphics Centurion XVI and significance was established at *p* < 0.05 for all comparisons.

## 3. Results and Discussion

### 3.1. Immunonutritional Bioactives

This study evaluates the modulatory role of the inclusion of *C. quinoa* and *S. hispanica* L. into *T. aestivum*-based bread formulations on the HFD-induced immunometabolic effects in conditions of caloric excess as well as hepatic steatosis, which is elicited by prenatal exposure to the obesogenic TBT [10]. Partial replacement of wheat flour by that from *C. quinoa* or *S. hispanica* L. into bread formulations was evaluated as an immunonutritional strategy to provide glycoside- and glucuronide-carrying proteins (2S seed storage protein), which have shown to display immunonutritional potential (Figure 2). Here, is shown the molecular weight of the molecular backbone as well as the glycoside (Figure 2A) and glucuronide (Figure 2B) linkage as deduced from the RP-HPLC-Ms/Ms [17] analyses on protein bands. These glycoside and glucuronide groups were not found to be associated with the wheat-derived immunonutritional bioactives [17]. Interestingly, the effects observed could not only be directly associated to these structural features but to their capacity to interact with TLR4 (discussed below). The inclusion of *C. quinoa* or *S. hispanica* L. flours modified the content of bioactive 2S globulins according to the following gradation: 2.21 ± 0.04 mg/g (WB) = 1.74 ± 0.32 mg/g (WQ) < 8.19 ± 0.11 mg/g (Ch). Focusing on the control of glucose homeostasis, the particular composition of the products help to provide additional information on how glycaemia results affected by the different carbohydrate and lipid changes (Table 1) [9].

### 3.2. Immunnutritonal Influence of Obesogenic Effects

To determine whether the different products affect hepatic lipid homeostasis, C57BL/6J mice were put on a 43% HFD (Figure 3), as established in previous studies [9]. Significant differences in the daily food (energy) intake between wild type or TBT-exposed mice in the different groups of treatment (30.0 kcal/day) (143.7 kJ/day) were not quantified. Feeding WB, only TBT-treated animals increased the body weight (BW) gain. However, contrasting patterns were observed for WQ and Ch (Figure 3A). Despite the model, animals receiving WQ increased BW, whereas those fed with Ch maintained a BW similar to that observed in the control group. These effects are attributed to bioactive ingredients in bread formulations, since it was previously shown that F1 animals directly exposed in utero to TBT display low effects on BW gain in eight-week-old animals [10]. Wild type and TBT-exposed animals fed HFD and administered with WQ showed a similar hepatosomatic (liver to BW) index of the animals compared to controls (Figure 3B). Otherwise, mice fed with WB and Ch showed higher liver/BW ratios. It is important to point out here that these variations were associated with changes in the liver weight for WB (1.30 ± 0.24 g), whereas in those administered with Ch (1.09 ± 0.13 g) and WQ (1.06 ± 0.05 g) it remained unaltered. As shown, iron deficiency favored higher values of the hepatosomatic index in all groups of treatment (Figure 3C), without causing differences in liver weight. In agreement with the TBT-derived obesogenic effects, animals coming from pregnancy females exposed to TBT displayed elevated triglycerides (TGs) levels (Figure 3D) that support alterations in hepatic lipid homeostasis. Plasmatic TGs levels in TBT-exposed animals ranged between 11.1 ± 6.6 and 45.8 ± 16.4%; higher than their respective counterparts. To further evaluate hepatic response(s), we quantified the relative variation of hepatic transcripts (mRNA) for lipogenic (i.e., FASN and SREBP1a) as well as coronary risk (i.e., ALOX15 and PTGS2) markers (Figure 3E–H). Rt-qPCR analyses of hepatic tissue revealed opposite patterns in the expression of FASN as a function of the treatment and feeding. In perinatal TBT-exposed mice, feeding WQ up-regulated FASN gene expression, whereas animals fed with WB and Ch exhibited down-regulation on this parameter. Data for SREBP1a showed similar trends for both models with significant higher expression levels in animals fed WQ. When considering the effects in the expression of ALOX15 and PTGS2, those were sharply increased in animals fed with WQ and Ch, but only slightly affected in those receiving WB. In this regard, it can be hypothesized that the administration of bread formulations containing *C. quinoa* or *S. hispanica* L. flour could help controlling the HFD/TBT-induced disturbances in lipid homeostasis. However, administration of WB seems to impair the physiological control of these biomarkers mostly in TBT-exposed animals.

Carbohydrates and polyunsaturated fatty acids (PUFA) are identified as positive [29] and negative [30], respectively, regulators of FASN transcripts. This dietary regulation fails to explain the changes observed in the FASN mRNA levels, and the apparent decrease of lipogenic markers in animals fed with WQ. TBT exposure raises the possibility of TBT-mediated alterations in molecules regulating glucose levels as well as fatty acid breakdown, for example, decreasing adiponectin [12,31]. FASN activity is required to promote cholesterol synthesis to facilitate TLR4 signal transduction and proinflammatory macrophage activation [32]. One of the most well-established transcription factors affecting FASN is SREBP; SREBP1 promotes fatty acid synthesis, while SREBP2 is more specific to cholesterol synthesis [33]. SREBP1 amplifies autophagy [34] that can regulate the hepatocellular lipid accumulation by its selective degradation. Accordingly, increased ALOX15 expression levels in animals fed WQ and Ch may suggest that mediators such as lipoxin, resolvin, and protectin, among other metabolites, could contribute to ameliorate hepatic inflammation and insulin resistance [35]. Furthermore, PTGS2 expression levels support protective effects against diet-induced steatosis by increased transcripts of hepatic cyclooxygenase (COX)-2 expression [36]. Collectively, these data suggest a potential increase of phagocytic conditions in animals fed with the different bread formulations (i.e., WQ >> Ch > WB) that may interpret the results from animals fed with WQ and Ch as a better control of the lipid accumulation.

### 3.3. Control of Alterations in Glucose Homeostasis

Hepatic lipogenesis results from insulin stimulation via lipogenic gene expression. To ascertain the possible association between changes in lipid homeostasis and alterations in insulin resistance, we measured plasmatic glucose concentrations and insulin levels to determine the HOMA-IR (Figure 4). This approach allowed to observe significant HOMA-IR increases in the HFD-groups administered with WB and Ch in relation to controls (Figure 4A). Feeding WQ and Ch products significantly reduced HOMA-IR relative to WB in perinatal TBT-exposed animals on a HFD. The effect on HOMA-IR values was similar, while inducing different trends in hyperglycaemic animals (streptozotocin-induced) fed with WQ and Ch on a HFD [9]. Glucose concentrations in both preclinical models were equal when feeding all bread formulations, showing increased values in relation to the control mice (Figure 4B). However, only animals fed with WQ and Ch displayed increased insulin levels. After removing the TBT stress, the animals could recover glucose homeostasis control via insulin receptor signal and insulin levels [12]. Here, both WQ and Ch bread formulations enabled a better control on insulin production (WQ > Ch >> WB) in perinatal TBT-exposed animals (Figure 4C). When comparing the effects of *C. quinoa*- and *S. hispanica* L-containing bread formulations on the relative variations in insulin production to ‘non-treated’ animals on the HFD, a similar reduction of insulin production in both hyperglycemic (streptozotocin-induced) [9] and perinatal TBT-exposed animals was observed. Taken together, variations in HOMA-IR seem to occur in TBT-exposed animals as an apparently improved insulin sensitivity (Figure 4B,C), as downward insulin levels seem to suggest. This is also supported by the absence of differences in postprandial glycaemia despite the significantly reduced carbohydrate content in Ch formulation, as well as the negligible effect of dietary polyunsaturated fatty acids from Ch in comparison to animals fed with WQ. Thus, changes in the immunonutritional bioactive protein fraction [9] seems to significantly take control on glucose homeostasis independently to the disease stage.

Because insulin resistance relates to intestinal epithelial TLR4 expression and activity [5], we examined whether the different bread formulations influenced glucose homeostasis in animals displaying a nutritional impairment of TLR4 signaling. Low iron levels selectively impair TLR4 signaling in macrophages through the TLR4/TRAM/TRIF pathway [37]. Research efforts identified the TRIF-dependent TLR signaling as a preventive factor in hepatic inflammation and diet-induced lipid accumulation [15,38]. Feeding with the different products, glucose concentrations in iron-deficient mice (ID, Haemoglobin, 11.1 ± 0.8 g/dL) were significantly reduced (Figure 4D): WB, 46%; WQ, 63% and Ch, 50% in relation to ID animals fed only with HFD, and WB, 52%; WQ, 68% and Ch, 58% to perinatal TBT-exposed mice. Animals receiving bread formulations exhibited significant lower BW values relative to those not receiving an additional feeding to the HFD, although this difference lacked statistical significance between the different products (Figure 4F). These observations transpire in the sense that dietary starch amount contributes but does not determine glycaemia, which seemed to depend, to a significant extent, on innate immune signals that stem at the intestinal level. Experimental data may allow to interpret a differential engagement of the TLR4/TRIF signaling by WQ and Ch (Figure 2), in line with a previous report [19]. Notably, the structural differences between bioactives provided by *C. quinoa* and *S. hispanica* L flours in comparison to wheat support their differential capacity to induce innate immunity via TLR4/MyD88-dependent signaling. This suggestion also aligns with the different FASN expression levels in TBT-exposed animals, as only TLR4/MyD88-dependent signaling impairs adiponectin signaling, contributing to insulin resistance [39].

### 3.4. Variations on Peripheral Immune Populations

The effects on immune cells and hepatocytes can modulate metabolism in T2D and obesity conditions and, reciprocally, the nutritional status influences immune homeostasis. The alterations in peripheral leukocyte populations in different animal groups are shown in Table 2 and Table 3. Wild type and TBT-exposed mice that were fed a HFD did not show differences in the RBC or WBC of the animals. WB administration caused downward trends in peripheral MY cells of wild type animals but significant increases in GR percentages in TBT-exposed animals. These alterations were not observed in those mice who were feed with WQ or Ch formulations. In contrast, upward trends in the MY population were observed through feeding with these samples. The variations relative to controls calculated in peripheral MY and LY populations in mice that were fed with the different products are shown in Figure 5. In animals without perinatal TBT exposure (Figure 5A), opposite significant variations in the MY population after feeding WB or WQ and Ch formulations were observed. These variations only reached statistical significance between animals fed WB and WQ. Meanwhile, variations in the LY population were no longer observed. By contrast, in perinatal TBT-exposed animals, MY variations were narrowed, losing statistical significance between the different groups of treatment (Figure 5B). Otherwise, feeding with WQ and Ch caused significantly increased LY in relation to WB. We do not have direct evidence to explain the increases in the LY population. In this sense, a possible explanation could be that the expansion of lymphoid cells and disturbances of hepatic glucose homeostasis may occur as sequential events derived from losses in metabolic control by myeloid cells. When considering iron-deficient mice, feeding both WQ and Ch products promoted clear increases in the MY population, which points out the significant influence of TLR4/TRIF signaling in these changes. However, innate immune signature in MY reached statistical significance only in animals fed with Ch, while changes in the LY population were not observed (Figure 5C).

Differences in immune stimulation can promote the modulation of cell-mediated immunity within the gut-liver axis, modulating the proportion of immune cells infiltrating into liver, and thereby HFD-induced insulin resistance. Hence, rt-qPCR analysis allowed to evaluate the relative mRNA levels of selected macrophage marker genes (Figure 5D–F). Since HFD-induced insulin resistance is commonly accompanied with chronic inflammation, the mRNA levels of CD68 (M1 phenotype) and CD206 (M2 phenotype) were measured (Figure 5D,E). In HFD-animals, the results show that feeding with WQ and Ch significantly down-regulated the CD68 gene expression in relation to WB. However, animals with perinatal TBT exposure displayed rather different results, showing a significant up-regulation of the CD68 transcripts in animals fed with WQ and Ch. In contrast, WB significantly decreased those transcripts in relation to their counterparts without TBT exposure. By comparison, CD206 transcripts exhibited similar trends but at a lower level than CD68. Experimental data show that both WB and Ch products up-regulate hepatic CD68 and CD206 mRNA levels, respectively, indicating that feeding does not block macrophage infiltration. These results allowed to calculate relative variations between macrophage’s M1/M2 phenotypes higher than those in controls (Figure 5F), which were not observed in those animals fed with Ch without perinatal TBT exposure.

Myeloid cells (i.e., monocytes and macrophages) dominantly express the TLR4 receptor in relation to lymphoid (i.e., ILCs and Th cells) populations. Taken together, data transpire in the sense of a peripheral myeloid trafficking/polarization as a key cell-mediated coordination process between hepatic metabolic lipid alterations and long-term, adaptive immune responses. This is in line with the generally accepted alterations in the macrophage metabolism contributing to impair liver dysfunction [40]. Increased myelopoiesis and M1 polarization may represent an integrated mechanism to control circulating nutritional fatty acids induced TLR4-mediated activation of macrophages [41] ending on and inhibited proliferation [14] and malfunction in the lympho-myeloid populations [42]. Furthermore, proliferation/survival and cytokine production of ILC2s, active contributors to diet-induced obesity [6], is suppressed by IFN-γ and, to a lesser extent, by IL-27, which expression in human macrophages occurs in an IFNα mediated fashion [43]. This could be an example of the important role of diet influencing reciprocal interaction between innate immunity and the metabolic system.

These data are consistent with previous studies demonstrating a role for innate lymphoid cells determining insulin secretion and glycaemia [6]. For example, alterations in the myeloid cell populations have been identified as potentially relevant effectors in the metabolic control of glucose homeostasis. However, the underlying innate immune signaling allowing myeloid control of liver metabolism is unclear. Because WQ and Ch provide immunonutritional agonists are able to interact with the innate immune TLR4, intestinal TLR4 may act as a key trigger of the immunometabolic processes. Notably, the immunonutritionally-induced resistance to obesity in animals developing a transgenerational increased liver fat accumulation, in which insulin resistance also occurs, was rescued by feeding with WQ and Ch but not with WB. These findings support that intestinal TLR4 signaling was likely important in the control of glycaemia and insulin resistance. Collectively, data evidenced a role for immunonutritional agonists from *C. quinoa* and *S. hispanica* in intestinal innate immunity in the control of glucose homeostasis.

## 4. Conclusions

In summary, the reciprocal contributions of lymphoid cells populations and lipid metabolism to immunonutritional outcomes of the products cannot be excluded. Indeed, lipid metabolism is important for the expansion and function of the lymphoid cells population, enhancing TLR4-mediated proinflammatory signaling [44]. Fatty acid synthase-dependent MyD88 palmitoylation is necessary for TLR4-induced inflammation [32]; however, both WQ and Ch may interact with TLR4 triggering MyD88-independent signaling in myeloid cells [19]. Thereby, WQ and Ch products improved glucose homeostasis in comparison to WB, this is consistent with previous studies demonstrating that it associates with early increased peripheral myeloid cells population [45,46]. Collectively, it is likely that immunonutritional agonists may contribute to maintaining the function of hepatic myeloid populations [i.e., monocytes and macrophages], which act as critical effectors determining the function of innate lymphoid cells (i.e., Tregs and ILCs) in the regulation of glucose homeostasis in mice on a HFD.

## Figures and Tables

**Figure 1 nutrients-13-01537-f001:**
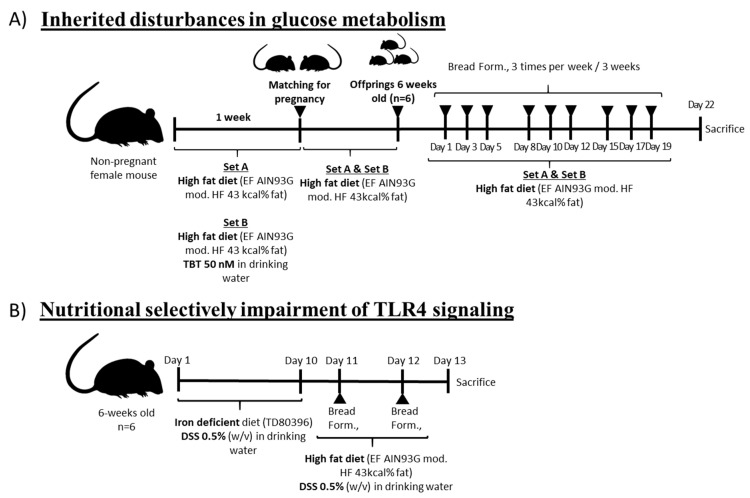
Schematic representation of the two experimental models used to mimic early disturbances in glucose homeostasis and the administration pattern for the bread formulations (bread form: wheat-, *Chenopodium quinoa*- and *Salvia hispanica*-based bread formulations. Model 1 (**A**): Male animals (n = 6/group) were split into two groups depending on their origin; (i) pregnant female mice under standard conditions (F0_A) or (ii) pregnant female mice exposed (F0_B) to the obesogenic tributyltin TBT 50 nM (*w*/*v*) via drinking water to develop a state of obesity [10]. After, both F1_A and F1_B generations were kept on a HFD until reaching 6 weeks of age. Model 2 (**B**): Male mice (n = 6/group, 6 weeks-old) from pregnant females under standard conditions were kept on an iron-deficient diet (AIN93G modified, Ssniff Spezialdiäten GmbH, Soest, Germany), exposed to 0.5%, and administered with dextran sulfate sodium (DSS) (*w*/*v*) via drinking water [27] for 10 days. After, animals were put on a HFD and continued to be exposed to 0.5% DSS for an additional 2 days.

**Figure 2 nutrients-13-01537-f002:**
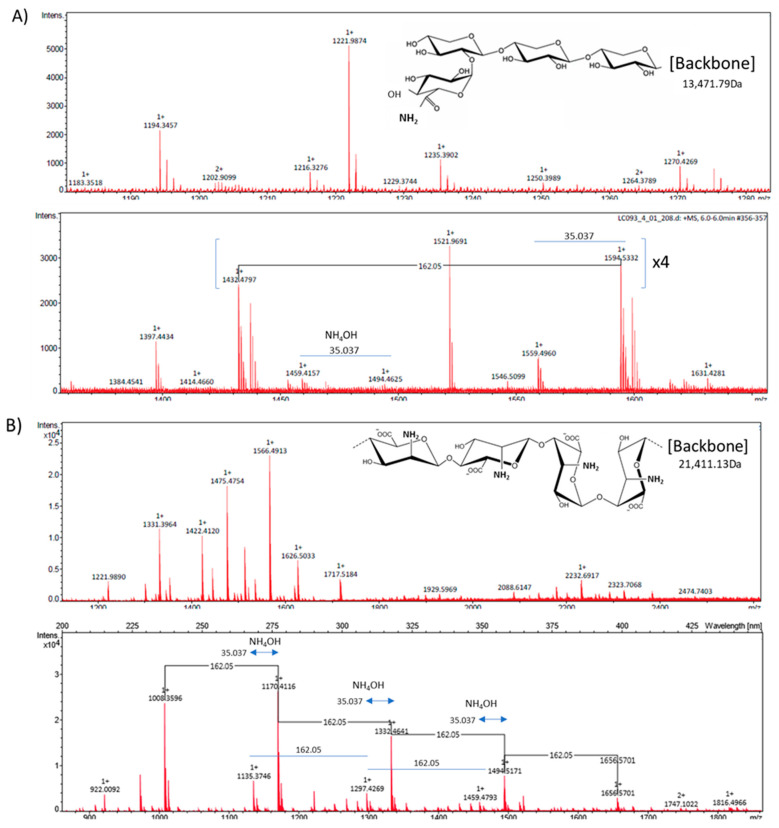
Mass spectrometry ‘*m*/*z*’ signals from the bioactive fraction obtained from *C. quinoa* (**A**) and *S. hispanica* L (**B**) [17,19].

**Figure 3 nutrients-13-01537-f003:**
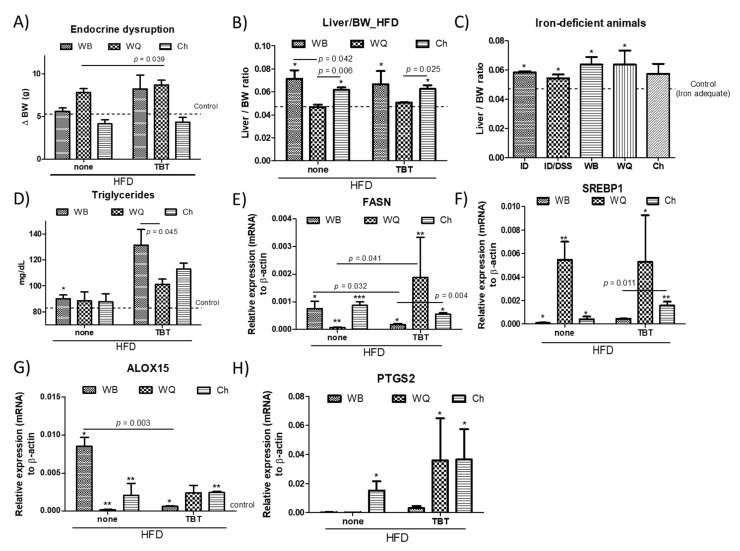
Effects of bread formulations-WB, wheat bread; WQ, white quinoa flour (25%)–containing bread; Ch, chia flour (20%)–containing bread-on body weight (BW) gain (**A**), changes in the hepatosomatic index (Liver/BW ratio) in wild type and tributyltin-exposed mice (**B**) as well as iron deficient mice (**C**), plasmatic triglyceride concentrations (**D**), and transcript levels (mRNA) of lipogenic markers-fatty acid synthase (FASN) (**E**) and sterol regulatory element-binding protein 1 (SREBP1a) (**F**), and coronary risk markers-arachidonate 15-lipoxygenase (ALOX15) (**G**) and prostaglandin-endoperoxide synthase (PTGS2) (**H**). Results are expressed as mean ± mean standard error (*n* = 3–6). Untreated controls are represented by the dotted line. *, **, *** indicates statistical differences between animals put under the same experimental model.

**Figure 4 nutrients-13-01537-f004:**
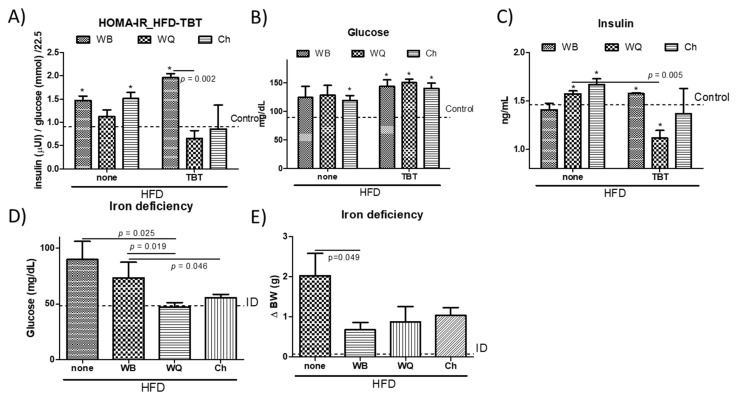
Biochemical parameters (HOMA-IR, **A**; glucose, **B** and insulin, **C**) in wild type and tributyltin (TBT)-exposed mice, glucose levels in iron-deficient (ID) (**D**) animals and body weight gain in ID animals (**E**), all fed with a high-fat diet and administered with different bread formulations: WB, wheat bread; WQ, white quinoa flour (25%)–containing bread and Ch, chia flour (20%)–containing bread. Results are expressed as mean ± mean standard error (*n* = 6). Untreated controls are represented by the dotted line. * indicates statistically significant (*p* < 0.05) differences in relation to controls.

**Figure 5 nutrients-13-01537-f005:**
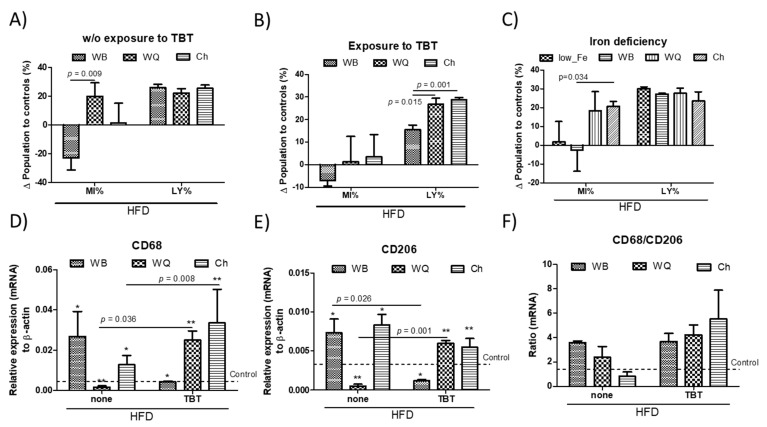
Peripheral relative variations of myeloid (MY) and lymphoid (LY) populations in wild type (**A**) and tributyltin (TBT)-exposed mice (**B**) as well as in iron-deficient (**C**) animals. Hepatic rt-qPCR analysis (mRNA) of selected macrophage marker genes; CD68 (**D**), CD206 (**E**) and their relative variation (**F**). Animals were fed with different bread formulations-WB, wheat bread; WQ, white quinoa flour (25%)–containing bread; Ch, chia flour (20%)–containing bread. Results are expressed as mean ± mean standard error (*n* = 6). Untreated controls are represented by the dotted line. *, ** indicates statistical differences between animals put under the same experimental model.

**Table 1 nutrients-13-01537-t001:** Proximal composition of *C. quinoa*- and *S. hispanica* L-containing bread formulations administered in the study.

Component g/100g d.m.		Bread	
	Wheat [23]	White Quinoa [23]	Chia [24,25]
Starch	66.2 ± 1.3	61.8 ± 1.7	53.3 ± 0.1
Proteins	12.4 ± 0.1	13.2 ± 2.5	14.1 ± 0.5
Lipids	1.1 ± 0.1	2.2 ± 0.1	7.8 ± 0.1
Ash	0.5 ± 0.1	1.5 ± 0.0	1.3 ± 0.0
Iron (µmol/g) [26]	0.4 ± 0.1	0.6 ± 0.0	0.7 ± 0.0
InsP6 (µmol/g) [26]	*n*.d.	2.0 ± 0.0	1.7 ± 0.0

Values are expressed as mean ± standard deviation (*n* = 3). d.m. dry matter; *n*.d. not detected; InsP_6_, phytic acid.

**Table 2 nutrients-13-01537-t002:** Total red blood cells (RBC) and leukocyte (WBC) counts, and lymphoid (LY), myeloid (MY), and granulocytes (GR) percentages (%) in peripheral blood of wild type animals and those perinatally-exposed to tributyltin. Wheat bread (WB), white quinoa bread (WQ) and chia bread (Ch). Results are expressed as mean ± standard deviation (*n* = 5–6). Different superscript letters (a–c) indicate statistical differences (*p* < 0.05).

Treatment		WB		WQ		Ch	
Parameter	Control	None	TBT	None	TBT	None	TBT
RBC (×10^9^/L)	8.8 ± 0.5 ^a^	8.5 ± 0.1 ^a^	8.6 ± 0.4 ^a^	8.7 ± 0.3 ^a^	8.6 ± 1.3 ^a^	7.6 ± 1.5 ^a^	7.4 ± 1.6 ^a^
WBC (×10^9^/L)	4.1 ± 0.9 ^a^	3.4 ± 1.1 ^a^	7.0 ± 1.2 ^b^	3.7 ± 1.1 ^a^	5.3 ± 1.2 ^a^	3.8 ± 1.1 ^a^	4.9 ± 2.8 ^a^
LY (%)	91.7 ± 5.3 ^a^	92.5 ± 3.5 ^a^	84.9 ± 5.1 ^a^	89.8 ± 4.6 ^a^	93.2 ± 2.9 ^a^	93.2 ± 3.5 ^a^	94.6 ± 1.6 ^a^
MY (%)	3.3 ± 1.5 ^a^	2.9 ± 1.1 ^a^	3.1 ± 0.3 ^a^	4.0 ± 0.8 ^a^	3.4 ± 1.4 ^a^	3.7 ± 1.3 ^a^	4.5 ± 1.7 ^a^
GR (%)	4.9 ± 1.8 ^a^	2.9 ± 1.1 ^a^	12.0 ± 4.8 ^b^	2.4 ± 1.9 ^a^	3.4 ± 1.6 ^a^	3.3 ± 1.6 ^a^	1.1 ± 0.7 ^c^

**Table 3 nutrients-13-01537-t003:** Total red blood cells (RBC) and leukocyte (WBC) counts, and lymphoid (LY), myeloid (MY), and granulocytes (GR) percentages (%) in peripheral blood of iron-deficient mice. Wheat bread (WB), white quinoa bread (WQ) and chia bread (Ch). Results are expressed as mean ± standard deviation (*n* = 5–6). Different superscript letters (a–c) indicate statistical differences (*p* < 0.05).

Treatment		WB	WQ	Ch
Parameter	Control			
RBC (×10^9^/L)	6.6 ± 0.8 ^a^	7.3 ± 0.8 ^a^	6.5 ± 0.9 ^a^	6.5 ± 0.7 ^a^
WBC (×10^9^/L)	2.3 ± 0.6 ^a^	3.1 ± 1.1 ^a^	3.4 ± 1.3 ^a^	3.6 ± 1.8 ^a^
LY (%)	95.9 ± 1.4 ^b^	93.5 ± 1.4 ^ab^	93.7 ± 3.8 ^ab^	90.8 ± 7.2 ^a^
MY (%)	3.8 ± 0.1 ^a^	3.3 ± 0.7 ^a^	4.0 ± 0.9 ^a^	4.0 ± 0.2 ^a^
GR (%)	2.9 ± 0.6 ^ab^	2.8 ± 1.5 ^ab^	1.5 ± 0.8 ^a^	3.8 ± 2.2 ^b^

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
