# Peer review of "Immunonutritional Bioactives from *Chenopodium quinoa* and *Salvia hispanica* L. Flour Positively Modulate Insulin Resistance and Preserve Alterations in Peripheral Myeloid Population"

_nutrients, 2021, doi:10.3390/nu13051537_

Round 1
Reviewer 1 Report
Overall, this is an interesting paper in order to study comparatively the immuno-nutritional effects of bioactives components from quinoa and chia flour replacing part of wheat flour in bread on animal model with disturbances in glucose metabolism and iron deficient diet.
The article is suitable to be published with minor revisions.
Page 3 Line 112 why the differences of 20 and 25% of chia and quinoa flour?
Page 3 line 129 Were the 3 formulations administered to each one of the two animal models used A) and B)? is not clear in your scheme.
Page 5 line 201 figure 2 needs to be explained in detail
Page 13 the conclusion is very well explained in the abstract, but it is not very clear at the end of the manuscript
What bioactive components, in addition to seed storage proteins, could be responsible for the results? Need a better explanation
Author Response
Reviewer 1
Overall, this is an interesting paper in order to study comparatively the immuno-nutritional effects of bioactives components from quinoa and chia flour replacing part of wheat flour in bread on animal model with disturbances in glucose metabolism and iron deficient diet.
The article is suitable to be published with minor revisions.
Page 3 Line 112 why the differences of 20 and 25% of chia and quinoa flour?
The reason to use different proportions of the S. hispanica and C. quinoa flour was to normalize the products in relation to their protein content as this fraction provides the immunonutritional bioactives (Ref. 9).
In view of the reviewer’s comment, a new sentence explaining the chosen percentages has been included (Page 4, Lines 115-116); “The different proportions of C. quinoa and S. hispanica flour were chosen to normalize the protein content in the products as immunonutritional bioactives derive from it”.
Page 3 line 129 Were the 3 formulations administered to each one of the two animal models used A) and B)? is not clear in your scheme.
As the reviewer correctly points, all 3 formulations were administered to each of the two animals presented in the study. The duration of the treatment varied due to the different objectives: 3 weeks to evaluate the influence on lipid homeostasis and 2 days to take into consideration the nutritional alteration of TLR4 signalling.
According to the reviewer’s comment an additional sentence clarifying the administration pattern has been included in the manuscript (page 5, line 133-135): ‘All bread formulations were administered (14 mg/day/animal) to the different animal models; three times per week for 3 weeks to model 1 (Fig. 1A) and for 2 consecutive days to model 2 (Fig. 1B)’. Also, it has been rewritten the legend for Figure 1 (page 6, line 137-139): ‘Schematic representation of the two experimental models used to mimic early disturbances in glucose homeostasis and the administration pattern for the bread formulations (bread form: wheat-, Chenopodium quinoa- and Salvia hispanica-based bread formulations’.
Page 5 line 201 figure 2 needs to be explained in detail
We feel sorry for this inconvenience. A major of our research line is the identification and characterization of the immunonutritional bioactives that were found in the protein fraction. To this end, we have performed RP-HPLC-Ms/Ms analyses on the protein bands isolated in the SDS-PAGE analysis (Ref. 17). In previous of our studies, it was determined that major differences between protease inhibitors found in wheat, C. quinoa and S. hispanica flour were the presence or not of glucoside/glucuronide prosthetic groups associated to these proteins (Ref. 17 and 19). These structures were concluded from the Ms/Ms analyses. Because of the nutritional nature of this journal, we considered adequate to show the Ms/Ms spectra and our conclusion. We are showing a representative spectrum of the analyses performed for each sample. In Ref. 17 it can be found the spectrum for wheat-derived immunonutritional bioactives without the carrying of glucoside-based groups (m/z, 162.05) and its association with NH4OH (m/z, 35.037) groups.
In the light of the reviewer’s comment some sentences to clarify these aspects have been introduced (page 8, line, 191-199): ‘Here, partial replacement of wheat flour by that from C. quinoa or S. hispanica L into bread formulations was evaluated as an immunonutritional strategy through the inclusion of glycoside- and glucuronide-carrying proteins (2S seed storage protein), which have shown immunonutritional potential (Figure 2). Here it is shown the molecular weight of the molecular backbone as well as the glycoside (Fig. 2A) and glucuronide (Fig 2. B) linkage deduced from the RP-HPLC-Ms/Ms [17] analyses on protein bands. These glycoside and glucuronide groups were not found associated to wheat-derived immunonutritional bioactive [17]. Interestingly, the effects observed could not only be directly associated to the proportion of these structural changes, but to their capacity to interact with TLR4 (discussed below).’
Page 13 the conclusion is very well explained in the abstract, but it is not very clear at the end of the manuscript.
Many thanks for your comment, which we really appreciate. In the light of the reviewer’s comment the paragraph summarizing the conclusions has been rewritten as follows (page 18, line 407-413): ‘Thereby, WQ and Ch products improved glucose homeostasis in comparison to WB, consistent with previous studies demonstrating that it associates with early increased peripheral myeloid cells population [41-42]. Collectively, it is likely that immunonutritional agonists may contribute to maintain the function of hepatic myeloid populations [i.e., monocytes and macrophages], which act as critical effectors determining the function of innate lymphoid cells (i.e., Tregs and ILCs) in the regulation of glucose homeostasis in mice under a HFD.’
What bioactive components, in addition to seed storage proteins, could be responsible for the results? Need a better explanation.
We really understand your comment. We attributed the effects to the 2S storage protein, based on in vitro (Plant Foods Hum Nutr. 2020, 75(4):592-598; Cells 2020, 9(3), 593.) and in vivo (Food Funct. 2020, 11(9):7994-8002) studies. In the latter, which conclusion is summarized in Ref. 9 (Food Funct. 2020, 11(9):7994-8002), it was discussed the potential effects derived from the carbohydrate and lipid fraction of bread formulations. These studies discussed the effects according to the chemical composition and allowed us to identify the protein fraction as unique promotor of the effects observed. The different bread formulations present differences in carbohydrates as well as in the content of lipids (wheat and C. quinoa vs S. hispanica). Cellular and organic response(s) could not be associated to the carbohydrate as well as the lipid content, for example in omega-3/6 components. However, the effects observed only appeared to be mostly attributed to the structural differences found between the immunonutritional bioactives from wheat and C. quinoa or S. hispanica, and not the lipid and carbohydrate fraction.
Reviewer 2 Report
The topics reported in the manuscript are each of interest. However, the complexity and confounding of the study design make following the studies a challenge. The conclusions based on the findings are not clear.
Specific Comments
- Grammar and sentence structure issues make it difficult to follow the background, reporting of results, and the interpretation of the findings.
- The abstract clearly stated the aim as “to compare the effects of immunonutritional bioactives from Chenopodium quinoa (WQ) or Salvia hispanica L (Ch) when used to partial replacement of wheat flour (WB) in bread formulations”. However, the experiment design is much more complicated: 1) transgeneration effects of high fat diet feeding; 2) obesogenic chemical exposure; and 3) modified carbohydrate feeding in the context of a high fat diet. The authors need to clarify the actual aim of the study. The background is hard to follow because it reflects the complexity of the design and not the straightforwardness of the aim stated in the abstract.
- The complexity of the manuscript is compounded by using an iron deficient model with DSS treatment but also high fat feeding and modulation of the carbohydrate type. The combination of iron deficiency with DSS treatment is complex enough that a study could be conducted just on that.
- Figure 4D has a STZ label. Were mice treated with STZ?
- Subheadings in the results/discussion would help in organizing the content so that it is easier to follow.
- The graphs are difficult to interpret: both stars and lines are used to show significance. Sticking with one system would be easier to follow. Some graphs also include lines with findings indicating p values above 0.05: inclusion of the lines for non-significant findings detracts from the findings.
- Given the large number of treatment groups, six animals per group is a low amount. Was a power analysis used to determine group size?
Author Response
Reviewer 2
The topics reported in the manuscript are each of interest. However, the complexity and confounding of the study design make following the studies a challenge. The conclusions based on the findings are not clear.
This study wanted to put together the fact that innate immunity determines the impact of alterations in lipid metabolism and C. quinoa and S. hispanica provide immunonutritional bioactives, which are bioaccessible within the food matrix, able to influence glucose homeostasis via interaction with innate immunity. The study is also intended to provide a molecular connection supported by previous of our own studies.
Specific Comments
Grammar and sentence structure issues make it difficult to follow the background, reporting of results, and the interpretation of the findings.
According to the reviewer’s comment the manuscript has been revised considering language issues.
The abstract clearly stated the aim as “to compare the effects of immunonutritional bioactives from Chenopodium quinoa (WQ) or Salvia hispanica L (Ch) when used to partial replacement of wheat flour (WB) in bread formulations”. However, the experiment design is much more complicated: 1) transgeneration effects of high fat diet feeding; 2) obesogenic chemical exposure; and 3) modified carbohydrate feeding in the context of a high fat diet. The authors need to clarify the actual aim of the study. The background is hard to follow because it reflects the complexity of the design and not the straightforwardness of the aim stated in the abstract.
We feel sorry for this misunderstanding. The objective of the study is ‘to compare the effects of immunonutritional bioactives from Chenopodium quinoa (WQ) or Salvia hispanica L (Ch) when used to partial replacement of wheat flour (WB) in bread formulations’. However, we need to reproduce major features of the disease mimicking in vivo situation. This is the reason to use different nutritionally driven models to reproduce the conditions leading to obesogenic conditions.
The complexity of the manuscript is compounded by using an iron deficient model with DSS treatment but also high fat feeding and modulation of the carbohydrate type. The combination of iron deficiency with DSS treatment is complex enough that a study could be conducted just on that.
The reason to use different models was to prove/evaluate the effectiveness of the immunonutritional bioactives from C. quinoa and S. hispanica taking into consideration major features where alterations of glucose and lipid homeostasis occur. Additionally, it was used an iron-deficient model with DSS due to the control that the micronutrient exerts on TLR4 downstream signalling (page 13, line 305-306).
Figure 4D has a STZ label. Were mice treated with STZ?
We feel sorry for this misunderstanding. Figure 4D displays a comparison between the insulin levels quantified in TBT-treated animals from the current study and hyperglycaemic animals (STZ-treated) from a previous study (Ref. 9).
In view of the reviewer’s comment, panel D in figure 4 has been eliminated and only discussed in the manuscript (page 12, line 283-286).
Subheadings in the results/discussion would help in organizing the content so that it is easier to follow.
According to the reviewers’ comment, it has been included subheadings throughout the discussion section.
The graphs are difficult to interpret: both stars and lines are used to show significance. Sticking with one system would be easier to follow. Some graphs also include lines with findings indicating p values above 0.05: inclusion of the lines for non-significant findings detracts from the findings.
We feel sorry for this inconvenience. Statistical comparisons have been performed in relation to i) controls that are represented by ‘*’, ii) within a same treatment that are represented as ‘*’, ‘**’ and ‘***’, and iii) between different treatments that are represented by lines showing p values. Due to the number of comparisons, we find difficult to use only one system for all comparisons.
Additionally, the reason to include lines showing p > 0.05 was that results are plotted as mean ± standard error. This could induce ‘visual’ errors in some comparisons due to the apparent difference (distance) between the mean values plotted.
Following the reviewer’s suggestion, lines with p>0.05 have been deleted from the figures.
Given the large number of treatment groups, six animals per group is a low amount. Was a power analysis used to determine group size?
The number of animals per group was initially calculated using the "PS power and sample size" software considering the size of the expected effect and the intrinsic variability based on previous studies.
Additionally, the studies were toughly adhered to the ‘3Rs’. Thus, the minimum number of animals – according to previous studies – was considered.
We completely agree with the reviewer that some parameters would need to include larger groups of animals. However, when considering altogether the batch of results the conclusions still can be considered valid: WQ and Ch products improved glucose homeostasis in comparison to WB in all models assayed.
Reviewer 3 Report
In this work the effects of breads based on Chenopodium quinoa and Salvia hispanica flour on glucose homeostasis and immune system were studied. The paper is interesting, however some aspects need to be explained. First, were the high-fat diets with the addition of wheat bread and quinoa and chia flours isocaloric?
Figure 3 shows the results of changes in body weight, liver to body weight ratio, TAG concentration, and changes in gene expression related to lipid metabolism for rats fed the HF diet with breads, and by TBT stimulated obesity or fed only high fat diets with breads. The changes in the ratio of liver to body weight for the second model - rats with iron deficiency were also presented.
On the other hand, the results for glucose, insulin and the HOMA-IR index for the first model, as well as the results of glucose level and changes in body weight for the model with iron deficiency are also presented later in the paper (Figure 4). Maybe it is worth considering whether to present these models separately? It would also be worth adding the results for the influence of the tested breads on the remaining parameters (including gene expression and TAG) for the model with iron deficiency. Additionally, in Figure 4E and 4F, some groups disappeared (HFD and ID/DSS).
The Figure 4D also shows the results for % insulin_STZ-TBT, according to the reviewer, this chart is not necessary because it relates to the results of an earlier work (on a different model) and slightly distorts the two research models presented. In order to examine the influence of the tested breads on insulin production, it is sufficient to explain this effect in the discussion, what has been done.
Please provide information about the Fe content in a deficit diet.
Please correct the abbreviation of Homeostatic Model Assessment of Insulin Resistance, using capital letters (HOMA-IR).
Author Response
Reviewer 3
In this work the effects of breads based on Chenopodium quinoa and Salvia hispanica flour on glucose homeostasis and immune system were studied. The paper is interesting; however some aspects need to be explained. First, were the high-fat diets with the addition of wheat bread and quinoa and chia flours isocaloric?
Many thanks for your comment, which it is really appreciated. We feel sorry because the manuscript does not state parameters such as food intake during the study period. In relation to this aspect, it has been included in the manuscript a new sentence stating, ‘There were no significant differences in daily food (energy) intake between the different groups of treatment (30.0 kcal/day) (143.7 kJ/day): C. quinoa = 2.76 ± 0.65 g/animal/day and S. hispanica = 2.43 ± 0.72 g/animal/day over the study period’. In this context, the caloric intake provided by 14 mg/day of the bread formulation is negligible in relation to that provided by the diet.
In the light of the reviewer’s comment, it has been introduced a new sentence stating the average caloric intake/animal/day (page 9, line 211-213).
Figure 3 shows the results of changes in body weight, liver to body weight ratio, TAG concentration, and changes in gene expression related to lipid metabolism for rats fed the HF diet with breads, and by TBT stimulated obesity or fed only high fat diets with breads. The changes in the ratio of liver to body weight for the second model - rats with iron deficiency were also presented.
On the other hand, the results for glucose, insulin and the HOMA-IR index for the first model, as well as the results of glucose level and changes in body weight for the model with iron deficiency are also presented later in the paper (Figure 4). Maybe it is worth considering whether to present these models separately? It would also be worth adding the results for the influence of the tested breads on the remaining parameters (including gene expression and TAG) for the model with iron deficiency. Additionally, in Figure 4E and 4F, some groups disappeared (HFD and ID/DSS).
The reason to show together data concerning both models is because HFD-fed animals and TBT-treated mice represent the chronic consequences of high caloric intake. Otherwise, the iron-deficient model was used to help a better understanding concerning the molecular downstream signalling associated to TLR4 that influences the organic effects observed. In the iron-deficient model gene expression and TAG were not measured due that iron repletion occurs quickly (2 days) during HFD-feeding. Thus, it could be difficult to identify what could be the cause, iron repletion or bread feeding, affecting these parameters. Therefore, it was decided to measure gene expression and TAG in the models after 3 weeks. In contrast, it was quantified glucose levels because TLR4 is essential regulating insulin resistance, and this parameter would result affected during the short time of feeding of the iron-deficient model.
In relation to Fig 4E and 4F, we feel sorry for the misunderstanding. Both figures correspond to the iron-deficient model.
In consideration of the reviewer’s comment, also according to the comments of other reviewers, subheadings have been included to facilitate the following and understanding of the discussion section and titles in Fig. 4E and 4F have been unified.
The Figure 4D also shows the results for % insulin_STZ-TBT, according to the reviewer, this chart is not necessary because it relates to the results of an earlier work (on a different model) and slightly distorts the two research models presented. In order to examine the influence of the tested breads on insulin production, it is sufficient to explain this effect in the discussion, what has been done.
We feel sorry for this inconvenience. Our idea was to highlight the differences of wheat-based bread feeding in the different preclinical models. Figure 4D displays a comparison between the insulin levels quantified in TBT-treated animals from the current study and hyperglycaemic animals (STZ-treated) from previous of our studies (Ref. 9).
In view of the reviewer’s comment, panel D in figure 4 has been eliminated, and it has only been discussed in the manuscript (page 12, line 283-286).
Please provide information about the Fe content in a deficit diet.
According to the manufacturer (https://insights.envigo.com/hubfs/resources/data-sheets/80396.pdf): ‘The diet contains approximately 2-6 ppm Fe. To limit background iron, cellulose is omitted, and reagent grade, pretested calcium phosphate is used in the mineral mix’.
Please correct the abbreviation of Homeostatic Model Assessment of Insulin Resistance, using capital letters (HOMA-IR).
Page 7, Lines 158-159. Many thanks for your comment. Our mistake has been corrected.
Round 2
Reviewer 2 Report
The authors have improved the readability of the manuscript and have been responsive to many of the points.
The readability issues in Figures 3-5 remain. Non-significant findings are still presented using lines between groups. The authors state that the comparisons are too complex to use only one system to identify significance. Thus, how were the comparisons made? Was a post hoc test used? If so, which one was used? The same lettering system used in Table 2 could also be used in the figures.
Author Response
The readability issues in Figures 3-5 remain. Non-significant findings are still presented using lines between groups. The authors state that the comparisons are too complex to use only one system to identify significance. Thus, how were the comparisons made? Was a post hoc test used? If so, which one was used? The same lettering system used in Table 2 could also be used in the figures.
In view of the reviewer’s comment, the non-significant findings have been deleted in all figures.
When saying that comparisons are complex, it means that it would be needed 4-5 letters per group to differentiate statistical variations between values within a same experimental model and in relation to the controls. Comparisons within a same experimental model were performed as already indicated in the manuscript (page 7, line 181-185). Otherwise, comparison between groups in different experimental models but receiving the same bread formulation was performed by ‘comparison of means’. The reason to use it was the different ‘physiological conditions’ of animals that were used for the studies. It was considered that this could be easily understood by the readers.
In view of the reviewer’s comment, the description of the statistical analysis has been fully explained as follows (page 7, line 181-187): ‘The results are presented as the mean and standard error of the mean (SEM). The statistical analysis between the different groups of treatment within a same experimental model was conducted using one-way analysis of variance (ANOVA) and the Kruskal–Wallis post hoc test by ranks. Because of the different conditions used to obtain the animals for the different models, the comparison between groups of animals receiving the same bread formulation was performed by comparison of means. Statistical analyses were performed with the software Statgraphics Centurion XVI and significance was established at P < 0.05 for all comparisons.’
Reviewer 3 Report
Most of the corrections have been made in the paper, but there are also some minor flaws.
In the iron-deficient model, in Figure 3B there were 5 groups of rats (ID, ID / DSS, WB, WQ and Ch), while in Figure 4D and 4E of these groups there were 4 (HFD, WB, WQ and Ch). Can the Authors explain and correct it?
In the description of Figure 3, the authors gave the sentence "* indicates statistical differences between the different groups receiving a same treatment...." does the phrase "the same treatment" refer to the model used ( HFD or HFD / TBT model) or the bread served in 2 groups of mice (eg. HFD +WB vs. HFD / TBT +WB)? These explanations are confusing and perhaps used inconsistently in the graphs presented (check figure 3B and 3D).
Please correct the description of Figure 4. One of the graphs was removed, but was still included in the description.
According to the previous suggestion, please change HOMAir to HOMA-IR (please write the abbreviation for insulin resistance in capital letters - IR).
Author Response
In the iron-deficient model, in Figure 3B there were 5 groups of rats (ID, ID / DSS, WB, WQ and Ch), while in Figure 4D and 4E of these groups there were 4 (HFD, WB, WQ and Ch). Can the Authors explain and correct it?
This figure shows changes in liver/BW considering the effect of iron deficiency (ID), inclusion of DSS (ID/DSS) corresponding to the model used, and the administration of the bread formulations to ID/DSS animals. All of those in comparison to control (untreated animals under an iron-adequate chow). The reason is to help the reader to understand the impact of the different variables on the model used, particularly, the inclusion of DSS. The reader can conclude that variations in liver/BW ratio are exclusively caused by iron deficiency, but not the inflammatory signals that could stem at intestinal level caused by the inclusion of DSS.
Later on, in other figures, we are showing only the data concerning the model used that is the relevant information for discussion: ID/DSS animals (model used) administered with the different bread formulations, and all compared to iron-deficient animals.
In the description of Figure 3, the authors gave the sentence "* indicates statistical differences between the different groups receiving a same treatment...." does the phrase "the same treatment" refer to the model used ( HFD or HFD / TBT model) or the bread served in 2 groups of mice (eg. HFD +WB vs. HFD / TBT +WB)? These explanations are confusing and perhaps used inconsistently in the graphs presented (check figure 3B and 3D).
We feel sorry for this misunderstanding. The term ‘treatment’ was used for animals under the same experimental model. Otherwise, ‘administration’ or ‘fed’ was used when referring to the bread formulations.
IN view of the reviewer’s comment the sentence has been rewritten as follows (page 11, line 249-250) *indicates statistical differences between animals put under the same experimental model.
Please correct the description of Figure 4. One of the graphs was removed, but was still included in the description.
Many thanks for your appreciation. The description of the aforementioned figure has been removed.
According to the previous suggestion, please change HOMAir to HOMA-IR (please write the abbreviation for insulin resistance in capital letters - IR).
The manuscript has been revised and unified the term HOMA-IR throughout.